# Effect of Heat Treatment on the Passive Film and Depassivation Behavior of Cr-Bearing Steel Reinforcement in an Alkaline Environment

Yuwan Tian, Cheng Wen *, Xiaohui Xi, Deyue Yang and Peichang Deng

Guangdong Provincial Ocean Equipment and Manufacturing Engineering Technology Research Center, School of Mechanical and Power Engineering, Guangdong Ocean University, Zhanjiang 524088, China
* Correspondence: wcheng.3jia@163.com

**Abstract:** Using Cr-bearing low-alloy steel is an effective preventive measure for marine structures, as it offers superior corrosion resistance when compared to plain carbon steel. However, it remains unclear how quenching and tempering heat treatment, which is commonly applied to steel reinforcement in some specific environments to improve its mechanical properties, affects its corrosion resistance. In the present work, the impact of heat treatment on the passive film and depassivation behavior of the 0.2C-1.4Mn-0.6Si-5Cr steel are studied. The results reveal that quenching and tempering result in grain refinement of the Cr-bearing steel, which increases its hardness. However, this refinement causes significant degradation in its corrosion resistance. The critical $[Cl^-]/[OH^-]$ ratio after quenching and tempering is determined to be approximately 6.6 times lower than that after normalization, and the corrosion rate is 1.6 times higher. After quenching and tempering, the passive film predominantly comprises iron oxides and hydroxides, with relatively high water content and defect density. Additionally, the $Fe^{II}/Fe^{III}$ ratio and film resistance are relatively low. In comparison, after normalization, the steel exhibits high corrosion resistance, with the passive film formed offering the highest level of protection.

**Keywords:** chromium-bearing steel; corrosion; grain size; heat treatment; passive film



## 1. Introduction

Chloride-induced corrosion of steel reinforcement precisely limits the service life of marine structures. The use of low-alloy steel reinforcement with higher corrosion resistance than plain carbon steel is an efficient preventative method in highly aggressive marine environments [1]. In recent years, Cr-bearing low-alloy steel has attracted increasing attention due to its superior corrosion resistance, mechanical properties, weldability, and low cost compared to traditional carbon steel and stainless steel.

A thin, protective passive film can be formed on steel reinforcement under the alkaline conditions of ordinary concrete. This film has a strong influence on the durability of the structure because in the chloride-contaminated concrete, the service life is usually assumed to be equal to the initiation timeframe when the breakdown of passive film occurs (Figure 1) [2]. Several studies indicate that the alloy element Cr is involved in the formation of passive films on Cr-bearing steel, as Cr facilitates the formation of $Cr_2O_3$ and $Cr(OH)_3$ and thus inhibits the further oxidation and hydration of passive films [3,4]. The modified passive film of Cr-bearing steel improves the corrosion resistance of steel: numerous studies have proven that the critical chloride content and resistance to pitting corrosion initiation are higher after Cr addition to carbon steel [5–9]. As a result, the passive film and its depassivation behavior for Cr-bearing steel play important roles in increasing corrosion resistance.

Generally, hot-rolled steel bars are used directly in practical applications. In some specific environments, such as prestressed concrete sleepers, heat-treated steel reinforcement

is applied to improve the mechanical properties of the materials used in terms of strength, ductility, yield ratio, and cold bending performance [10]. Heat treatment can lead to a change in the microstructure of steels in terms of the carbides, grain size, and so on. Numerous works have observed a direct relationship between localized corrosion susceptibility and the presence of carbide phases in steel [11,12], since carbides with a noble electrochemical equilibrium potential accelerate galvanic corrosion and ferrite dissolution [13,14]. Even worse, carbides such as cementite can continuously aggravate galvanic corrosion during corrosion evolution processes due to the sustained accumulation of cementite on the steel surface, which can lead to an increase in the surface area ratio between the cathode and anode [12,15]. However, a recent study reported that an increase in the carbide content could slow the corrosion rate for steel, as it can improve the formation kinetics for passive films [16]. The grain size also has a considerable effect on the corrosion process [17–20]. Wang et al. noted that the relationship between the grain size and corrosion rate depends on the corrosive medium [21]. In an active corrosion system, grain refinement can accelerate corrosion due to the high electrochemical activity of grain boundaries, while in a passive solution, grain refinement can lead to an improvement in corrosion resistance [21–23]. In addition, some researchers have suggested the importance of structural heterogeneity (simultaneous presence of small and large grains) in decreasing the corrosion resistance of steel, and they observed that the corrosion rate is more dependent on grain size and the corresponding dispersion of grains with different grades [24].

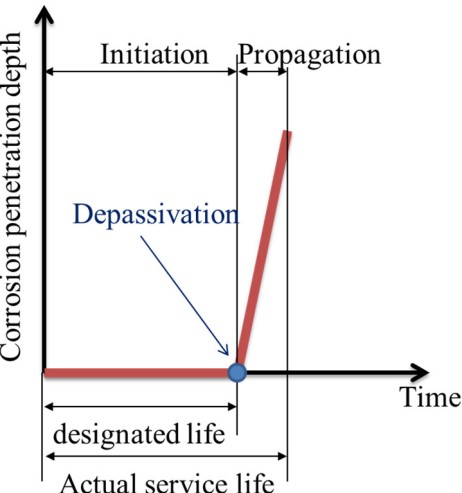

**Figure 1.** The service life including the initiation and propagation periods for corrosion in a reinforced concrete structure (Tuutti's model) [2].

However, the effect of heat treatment on the passivation and depassivation behavior of Cr-bearing low-alloy steel remains unclear. Does heat treatment influence the protective ability of passive film and thus affect the structural durability? How does heat treatment influence the property of passive film, that is, the Cr distribution, chemical composition, structure, and electrical characteristics in the passive film? In fact, previous studies have established the mechanism of effect of heat treatment on corrosion behavior of plain steel, but few studies are available regarding the use of different heat treatment processes for Cr-bearing steel reinforcement.

In this study, three types of heat treatment, namely, annealing, normalizing, and quenching and tempering, were applied to the as-received 5 wt.% Cr steel (0.2C-1.4Mn-0.6Si-5Cr). Then, electrochemical measurements, immersion tests, and physical characterization techniques were employed to investigate the effect of heat treatment on the microstructure evolution, passivation, and corrosion behavior of Cr-bearing steel in a simulated marine concrete environment.

## 2. Experimental Procedures

In order to evaluate the effect of heat treatment on the corrosion resistance of Cr-bearing steel, different heat treatments were carried out on the steel specimens, followed by physical characterization of the surface passive film, and finally the corrosion rate was detected. The specific workflow is shown in Figure 2.

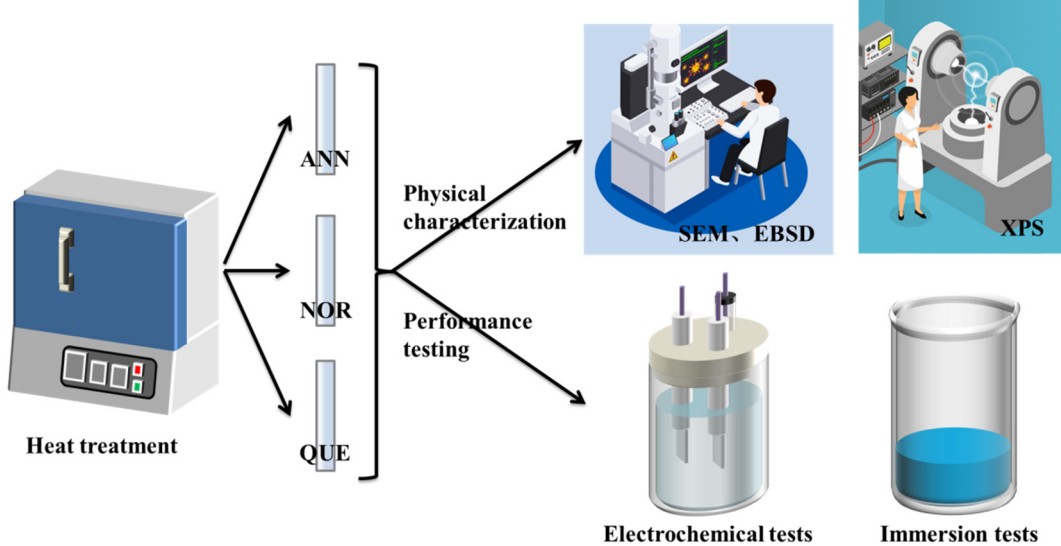

**Figure 2.** The workflow of the present work.

### 2.1. Material and Heat Treatment

The Cr-bearing steel reinforcement used in the present study consisted of a 400 MPa-grade hot-rolled ribbed bar, and its chemical composition (wt.%) was 0.19 C, 0.60 Si, 1.39 Mn, 0.02 P, 0.09 S, 5.20 Cr, and Fe for balance. The Cr-bearing steel rebar was machined into sheets with a size of 20 mm × 20 mm × 10 mm; ground with SiC emery paper; cleaned with saturated calcium hydroxide, water, and alcohol; and dried with nitrogen.

Three different heat treatment routes without a controlled atmosphere were adopted. The steel rebar specimens were kept at 900 °C for 40 min, and then cooled to room temperature in a furnace, air, and water and labelled ANN, NOR, and QUE, respectively. The QUE specimens were then tempered at 480 °C for 20 min and air-cooled to room temperature.

### 2.2. Microstructure

The heat-treated samples were then mechanically ground with SiC paper, polished with Ar ions, and etched with 4 vol.% nitric alcohol. Optical microscopy (Leica Microsystems, DVM6, Würzburg, Germany, and accuracy of 1 μm) and field emission scanning electron microscopy (TESCAN SEM, CLARA-GMH, Brno, Czech Republic, and accuracy of 0.8 nm) were utilized to observe the microstructures of Cr-bearing steels. The grain distribution and orientation were observed by electron backscattering diffraction analysis (Oxford Instruments EBSD, Symmetry, Oxford, UK, and accuracy of 0.1°). The crystal phase composition was examined in the angular range of 5–90° with X-ray diffraction (Bruker XRD equipment, D8 DISCOVER, Canton, USA, and accuracy of 0.001°) equipment with a Cu target.

### 2.3. Passive Film

X-ray photoelectron spectroscopy (Thermo Fisher XPS equipment, Thermo ESCALAB 250Xi, Wilmington, USA, and energy resolution of 15 meV) was used to characterize the Cr-bearing steel passive film after immersion in saturated calcium hydroxide, and all XPS spectra were calibrated by assuming a $C_{1s}$ peak at 284.8 eV.

Electrochemical tests (Autolab electrochemical workstation, 302 N, Utrecht, Netherlands, voltage accuracy of 0.1 mV, and current accuracy of 10 nA) including impedance spectroscopy (EIS) results and Mott-Schottky (MS) curves for the Cr-bearing steel were analyzed to characterize the properties of the passive film. A classic three-electrode cell in simulated concrete pore solution was used, with a Pt foil counter electrode, a Cr-bearing steel working electrode, and an SCE reference electrode. The steel rebar was machined into dimensions of 10 mm $\times$ 10 mm $\times$ 5 mm, welded with copper wire, embedded in epoxy resin with a working surface of 1 cm$^2$, and ground with #2000 emery papers. EIS was measured from 100 kHz to 10 mHz at the open-circuit potential with a sinusoidal potential perturbation of 10 mV. MS was tested from 1 to $-1$ $V_{SCE}$ with a step rate of 50 mV/s using a sinusoidal signal with a frequency of 1 kHz and an amplitude of 10 mV. All electrochemical measurements were repeated at least three times to ensure the repeatability of the data.

### 2.4. Depassivation

To determine the critical concentration of the chloride threshold ($C_{crit}$), the open-circuit potential (OCP) was continuously recorded for Cr-bearing steel in saturated calcium hydroxide with the gradual addition of 0.05 M chloride ion per 24 h. When the OCP displayed a sharp shift to a negative value, the corresponding value for the chloride concentration was defined as $C_{crit}$.

The corrosion behavior of Cr-bearing steel was also studied using potentiodynamic polarization measurements from $-300$ $mV_{OCP}$ to 700 $mV_{SCE}$ at a sweep rate of 0.1667 mV/s. The working electrode was immersed for 2 h to satisfy the stability requirements for an open-circuit potential (OCP) fluctuation lower than 5 mV/10 min. The electrolyte was composed of saturated $Ca(OH)_2$ + $NaHCO_3$ + $Na_2CO_3$ + 1.25 M NaCl (pH 11) to simulate aggressive concrete due to carbonation or penetration of salts.

## 3. Results

### 3.1. Microstructure and Phase Distribution

Figure 3 shows the metallographic structure for Cr-bearing steels with different heat treatments. The microstructure of the ANN sample is mainly ferrite and contains a minor amount of pearlite. Normalization increases the pearlite content and further refines the grain size of Cr-bearing steel. The Vickers hardness of ANN and NOR is low, with values of 110 HV and 161 HV, respectively. For the QUE sample, the microstructure presents less clear characteristics for lamellar pearlite, and the steel hardness increases to 249 HV.

Figure 4 shows the EBSD results obtained for Cr-bearing steels with different heat treatments. All the samples show a uniform microstructure. The major differences in grain size and grain boundary density can be observed, while the misorientation angle and its distribution are similar for all the samples.

### 3.2. Passive Film

Figure 5 shows the XPS spectra measured for the passive film on the Cr-bearing steels in saturated $Ca(OH)_2$ solution without any aggressive ions. The major peaks of O, Cr, Fe, C, and Ca can be observed in the survey spectra measured for the different surface films. The C and Ca elements are not considered because their presence may be attributed to the contamination of the immersion solution and the surrounding environment. Thus, the passive films are mainly composed of oxides and hydroxides of Fe and Cr. These conclusions are consistent with those in the literature [25–27], in which a bilayer structure with an outer Fe$^{III}$-enriched layer and an inner protective layer composed of anhydrous mixed Fe-Cr oxide have been reported. Since the XPS element sensitivity factors for Cr and Fe are very similar, the ratio of Cr oxides/Fe oxides in the passive film can be roughly assessed based on the peak intensity observed in Figure 5a. Clearly, the content of chromium oxides in the passive film is higher than that of the iron species for ANN and NOR, whereas iron oxides are predominant in the QUE passive film. The chromium in the oxide film has been confirmed to play a positive role in corrosion resistance since Cr

can act as a barrier to the transfer of electrons and holes in the space-charge layer [28,29], and Cr oxides exhibit a compact spinel structure. This indicates that the passive films on ANN and NOR are more stable and protective than those on QUE, which will influence the electrochemical properties and corrosion resistance of steel in corrosive media.

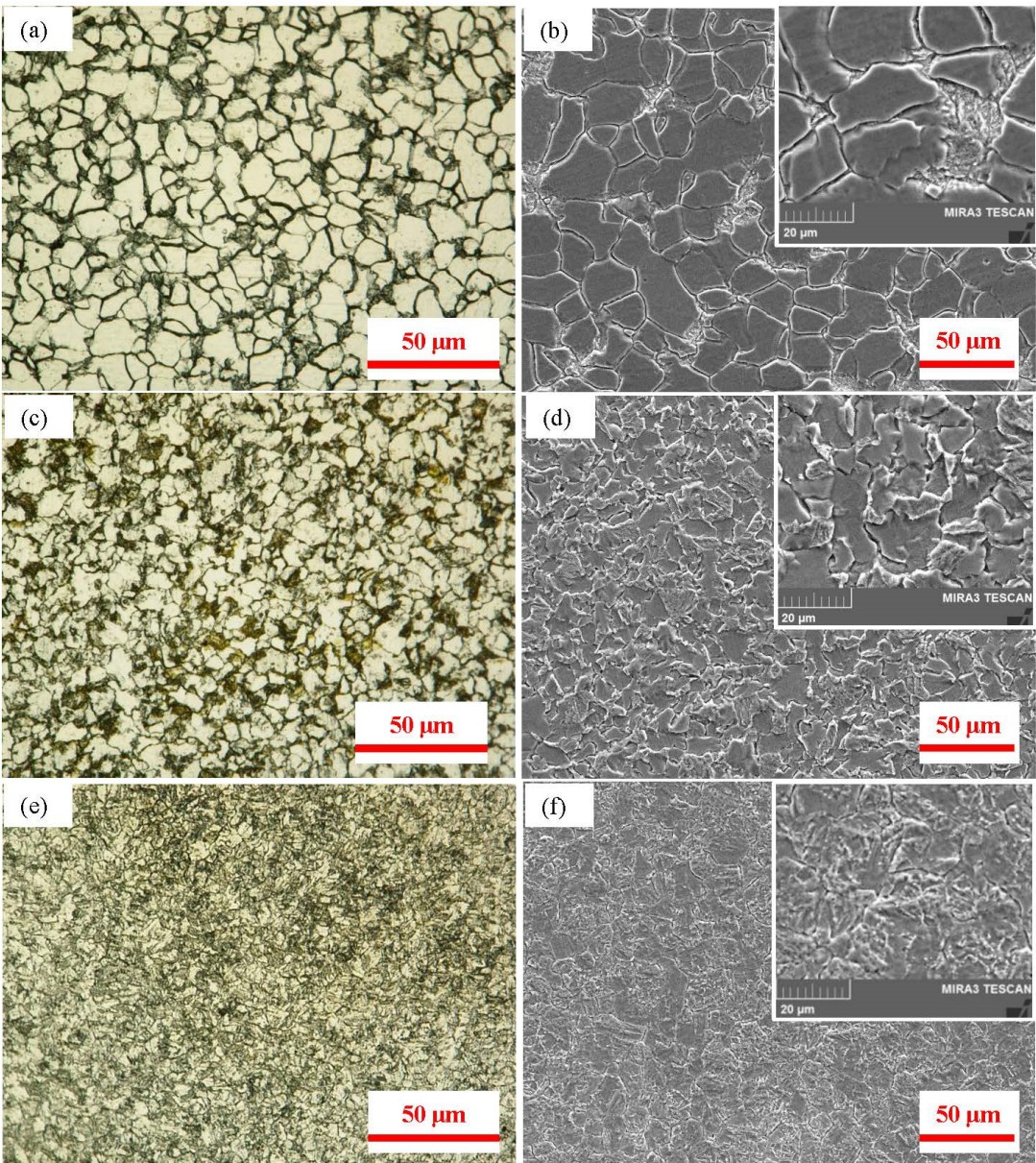

**Figure 3.** Microstructures for Cr-bearing steel with different heat treatments using optical and scanning electron microscopy: (**a**,**b**) ANN, (**c**,**d**) NOR, (**e**,**f**) QUE.

The typical Fe $2p_{3/2}$ and O 1s high-resolution XPS spectra shown in Figure 5c,d consist of different components, for which the corresponding binding energies and full widths at half maxima (FWHM) are listed in Table 1 [30–34]. All peaks were calibrated to the hydrocarbon (C 1s) signal set at 284.8 eV. The Shirley background correction algorithm was used to determine the optimal spectral baseline and peaks. The Fe 2p spectrum consists of a doublet structure of Fe $2p_{3/2}$ and Fe $2p_{1/2}$, and only the Fe $2p_{3/2}$ with a higher intensity was used in the present analysis. Quantitative composition information was determined using the corresponding area for each peak. From the Fe $2p_{3/2}$ high resolution spectra measured for ANN, NOR, and QUE, the proportion of Fe metal is determined to be 39%, 44%, and 25%, and the relative water content is 23%, 7%, and 23%, respectively. Since the

experimental environment and parameters are consistent for all tests, the effective volume for the XPS detection should likewise remain relatively constant. Thus, the passive film on NOR is the thinnest but the most dehydrated, while QUE displays the thickest passive film with a higher content of bound water. This is reasonable since there are two stages in the formation process of passive films [35]: the formation of a protective anhydrous $Fe^{II}$ oxide layer directly on the top of the metal substrate and the formation of a nonprotective hydrous $Fe^{III}$ layer due to the thickening and hydration of the $Fe^{II}$ layer. Thus, the high water content in the passive film of the QUE increases the electrochemical activity and accelerates the growth of the outer film, which results in a thicker but less protective oxide film.

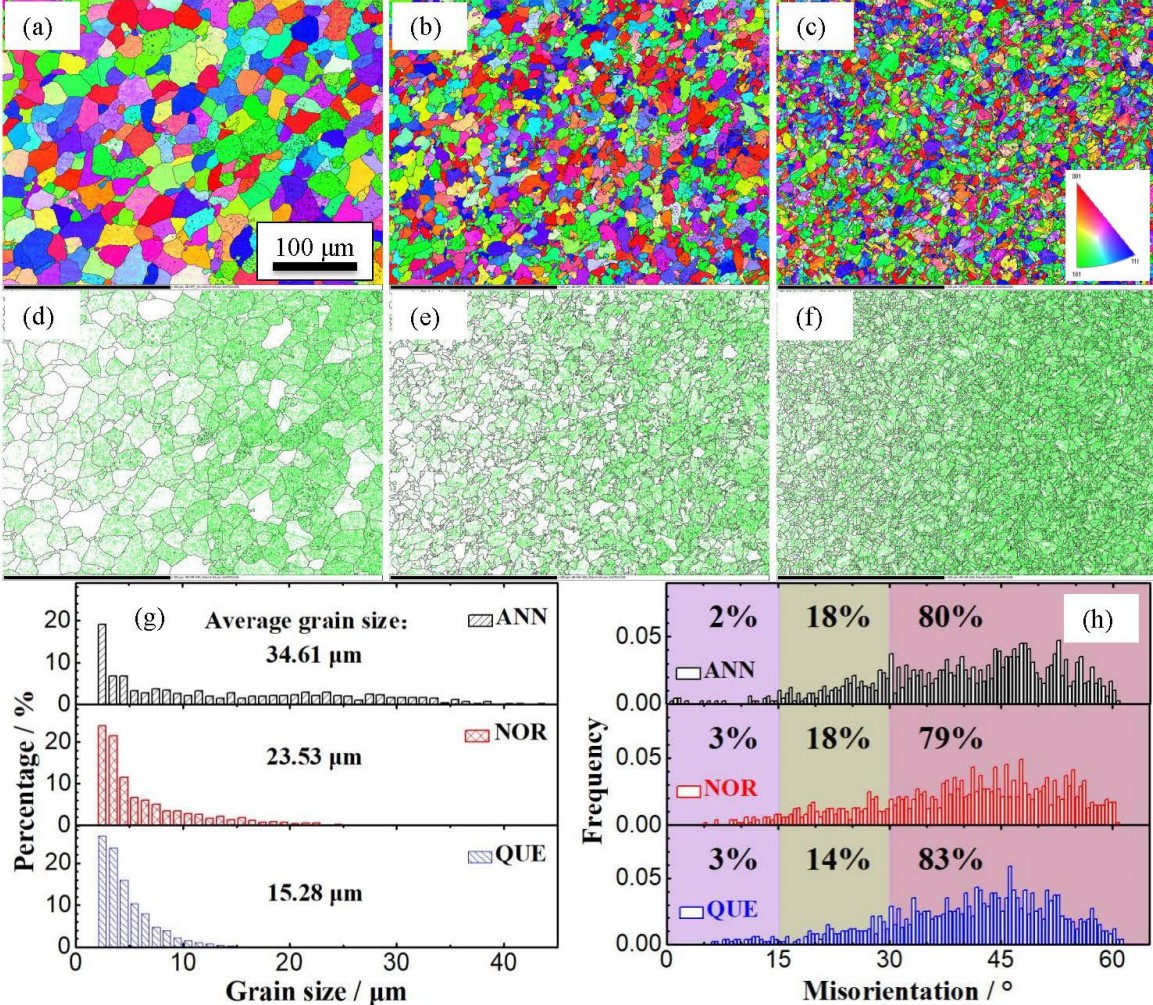

**Figure 4.** EBSD results for Cr-bearing steel with different heat treatments using scanning electron microscopy equipped with electron backscattered diffraction: the inverse pole figure (IPF) maps and corresponding grain boundary maps of ANN (**a**,**d**), NOR (**b**,**e**), and QUE (**c**,**f**); the black lines stand for the high-angle grain boundaries (HAGBs) and the green lines for the low-angle grain boundaries (LAGBs); (**g**) the distribution of grain sizes; (**h**) the distribution of misorientation angles.

The percentage contents for different iron oxides and hydroxides in the passive films are illustrated in Figure 5e. The $Fe^{II}$ oxide fraction is approximately 41% for ANN and 49% for NOR, while it is only 27% for QUE, which implies a much thicker inner $Fe^{II}$ layer for the passive film on NOR. These results are consistent with the presence of the thinnest passive film on NOR, as the thick $Fe^{II}$ layer inhibits the growth of the outer $Fe^{III}$ layer and thus decreases the total thickness of the passive film.

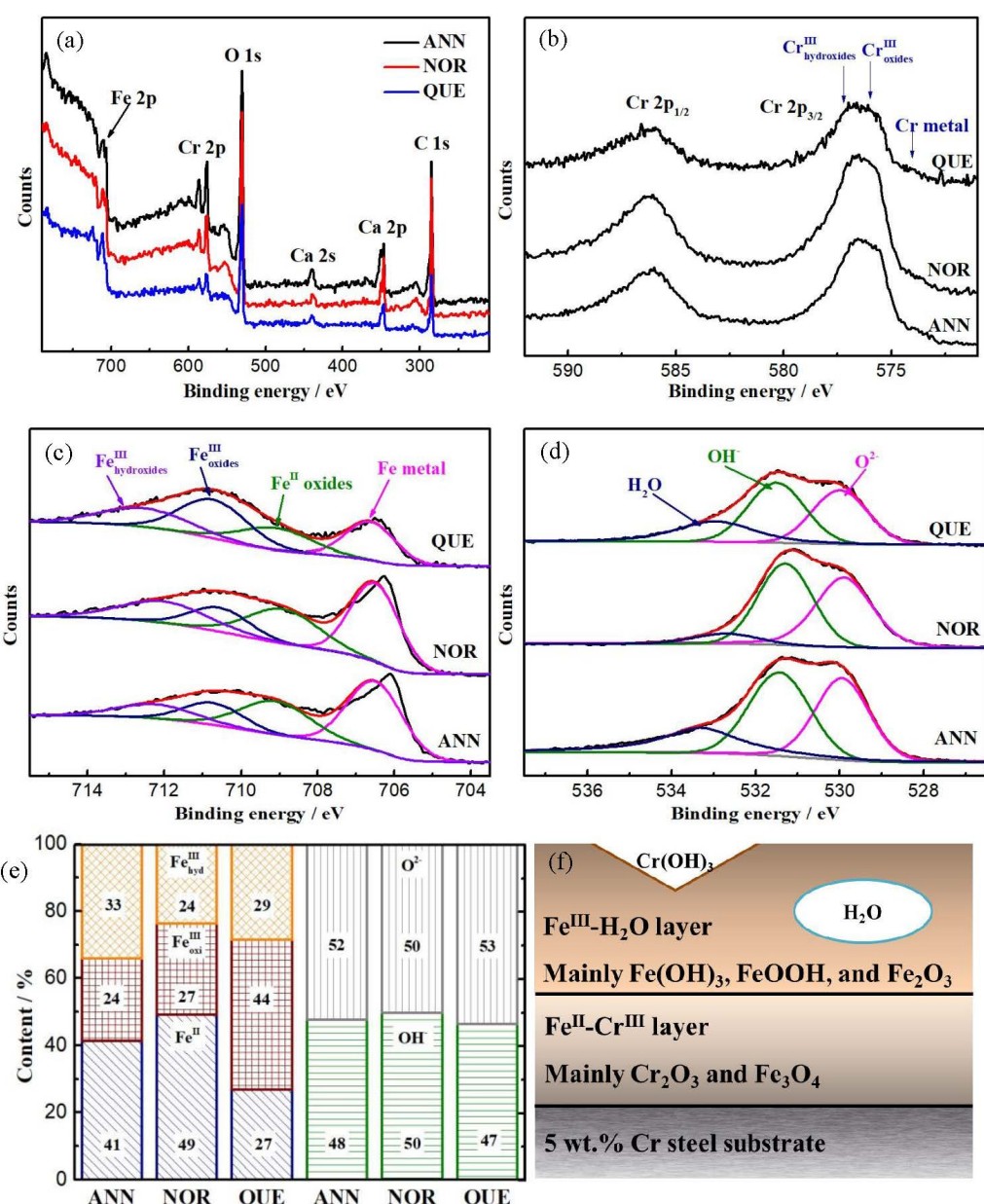

**Figure 5.** (**a**) XPS survey spectra of the passive film on Cr-bearing steel in the saturated Ca(OH)$_2$ solution. (**b**) High-resolution XPS spectra of Cr 2p. (**c**) High-resolution XPS spectra of Fe 2p$_{3/2}$. (**d**) High-resolution XPS spectra of O 1 s. (**e**) Fractions of the iron oxides and hydroxides, (**f**) A model of the passive film on Cr-bearing steel.

**Table 1.** The peak parameters of Fe 2p and O 2p XPS spectrum used in the present work.

| | Assignment | Fe Metal | Fe$^{II}$ | Fe$^{III}$ in Oxides | Fe$^{III}$ in Hydroxides |
|---|---|---|---|---|---|
| Fe 2p | Binding energy/eV | 706.9 ± 0.3 | 709.2 ± 0.3 | 711.0 ± 0.3 | 712.7 ± 0.3 |
| | FWHM/eV | 1.5 ± 0.2 | 2 ± 0.2 | 2 ± 0.2 | 2.5 ± 0.2 |
| | Assignment | O$^{2-}$ | OH$^-$ | H$_2$O | – |
| O 1s | Binding energy/eV | 530.2 ± 0.3 | 531.4 ± 0.3 | 533 ± 0.3 | – |
| | FWHM/eV | 1.5 ± 0.2 | 1.5 ± 0.2 | 2 ± 0.2 | – |

In addition, all three types of samples display an O$^{2-}$/OH$^-$ ratio that is almost equal to 1, which conflicts with the analyses for the Fe$^{II}$ content and Cr/Fe ratio in the passive film.

Based on the literature, hydroxides are supposed to be mainly enriched in the outer layer and represented in the forms of Fe(OH)3, FeOOH, and Cr(OH)3, as XPS depth profiling analyses show that OH- exhibits a declining tendency with increasing distance from the free surface of the passive film and becomes more difficult to detect at depths greater than several nanometers [3,36]. As a result, at least theoretically, the $O^{2-}$/$OH^-$ ratio should follow the order of NOR > ANN > QUE. One possible reason for this phenomenon is the contamination of Ca(OH)2.

A model for the distribution of different oxidation states in the passive film is illustrated in Figure 5f. In terms of the overall performance, the passive film on NOR can be considered the most protective compared to that for ANN and QUE. Even though the total thickness of the passive film on NOR is very thin, it exhibits the best resistance to aggressive ions, due to the barrier effect of the thickest inner $Fe^{II}$ layer, electrochemical inactivation due to the low water content, and modification due to the high Cr/Fe ratio.

Figure 6 shows the Mott-Schottky curves for Cr-bearing steels in a saturated Ca(OH)$_2$ solution without any aggressive ions. The semiconductive behavior of the passive film on the steel can be assessed according to the Mott-Schottky relationship [37]:

$$\frac{1}{C^2} = \frac{1}{C_H^2} + \frac{2}{\varepsilon \varepsilon_0 e N_q}\left(E - E_{FB} - \frac{kT}{e}\right)$$

where $C_H$ is the Helmholtz capacitance (22 $\mu$Fcm$^2$), $\varepsilon$ is the dielectric constant (12), $\varepsilon_0$ is the vacuum permittivity ($8.854 \times 10^{-12}$ F/m), $e$ is the elementary charge ($1.602 \times 10^{-19}$ C), $N_q$ is the carrier concentration, $E$ is the applied potential, $E_{FB}$ is the flatband potential, $k$ is the Boltzmann constant ($1.381 \times 10^{-23}$ J/K), and $T$ is the absolute temperature. Table 2 and Figure 6b show the calculated parameters for the electronic properties of the passive films, and detailed information for the formulas is provided in the literature [37].

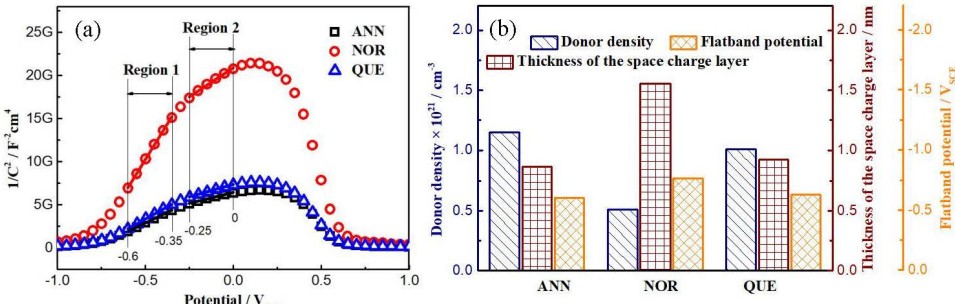

**Figure 6.** (**a**) Mott-Schottky curves for Cr-bearing steel in the saturated Ca(OH)$_2$ solution. (**b**) Electronic properties of the passive films.

**Table 2.** Fitting results for the Mott-Schottky plots.

| Sample | Slope$_1$ | Slope$_2$ | Donor Density, cm$^{-3}$ | Flatband Potential, mV$_{SCE}$ | Thickness, nm |
|---|---|---|---|---|---|
| ANN | $5.02 \pm 0.09 \times 10^9$ | $9.87 \pm 0.04 \times 10^9$ | $1.15 \pm 0.01 \times 10^{21}$ | $-608 \pm 3$ | $0.869 \pm 0.08$ |
| NOR | $13.5 \pm 0.03 \times 10^9$ | $33.0 \pm 0.04 \times 10^9$ | $0.51 \pm 0.01 \times 10^{21}$ | $-773 \pm 8$ | $1.553 \pm 0.03$ |
| QUE | $5.74 \pm 0.10 \times 10^9$ | $11.3 \pm 0.11 \times 10^9$ | $1.00 \pm 0.06 \times 10^{21}$ | $-638 \pm 17$ | $0.926 \pm 0.09$ |

The passive films on all three samples exhibit n-type semiconductive behavior, and the charge carriers are mainly in the donor state, as evidenced by the positive slopes observed in the swept potential range. The two separate linear regions indicate two different donor states, with $Fe^{II}$ donors being oxidized from tetrahedral and octahedral sites in the oxide crystal lattice [37]. The $Fe^{II}$ donors at the tetrahedral sites are more important in the passivation/depassivation processes because they are more readily excited at low potential and room temperature. The $Fe^{II}$ donor density at tetrahedral sites of NOR is significantly lower, down to approximately half that of ANN and QUE. It is slightly higher for ANN than

for QUE but remains on the order of $10^{21}$ cm$^{-3}$. The opposite trend for the space-charge layer thicknesses can be observed, i.e., the donor density displays an inverse relation with the thickness of the space-charge layer. The thickness of the space-charge layer for ANN, NOR, and QUE is 0.87, 1.55, and 0.93 nm, respectively, which is within the thickness range of the protective Fe$^{II}$ layer of the passive film [38]. This is a reasonable result since the oxidation and hydration processes for the inner Fe$^{II}$ layer to the outer Fe$^{III}$ layer will lead to the injection of numerous impurities into the film together with a thinning of the inner protective layer. Moreover, the thickness of the space-charge layer is in accordance with the XPS results for the Fe$^{II}$ content. Evidently, the passive film on NOR is less disordered, and the inner protective layer is thicker when compared with the layers on ANN and QUE. The flatband potential is much more negative for NOR than for ANN and QUE, and the difference can reach as high as 0.15 V, revealing the different states of the electrode and the electrolyte.

Figure 7 shows the electrochemical impedance spectroscopy for Cr-bearing steels in the saturated Ca(OH)$_2$ solution without any aggressive ions. The equivalent circuit containing two time constants in parallel [39] is used to describe the interfaces between the working electrode and electrolyte, as shown in Figure 7c, in which $R_s$ represents the solution resistance, $Q_{dl}$ is the electric double-layer capacitance, $R_{dl}$ is the charge transfer resistance, $Q_f$ is the film capacitance, $R_f$ is the film resistance, and $R_p$ is the polarization resistance. The most widely used equivalent circuit with two hierarchical parallel RC loops in series [40] is not used here due to the fine protective efficiency of the passive film in highly alkaline media and also due to the fact that this equivalent circuit gives less fitting parameters in the present work. The impedance function of the $R(QR)(QR)$ model [41] is:

$$Z = R_s + \frac{R_{dl}}{1 + (j\omega)^\alpha R_{dl}Q_{dl}} + \frac{R_f}{1 + (j\omega)^\alpha R_f Q_f}$$

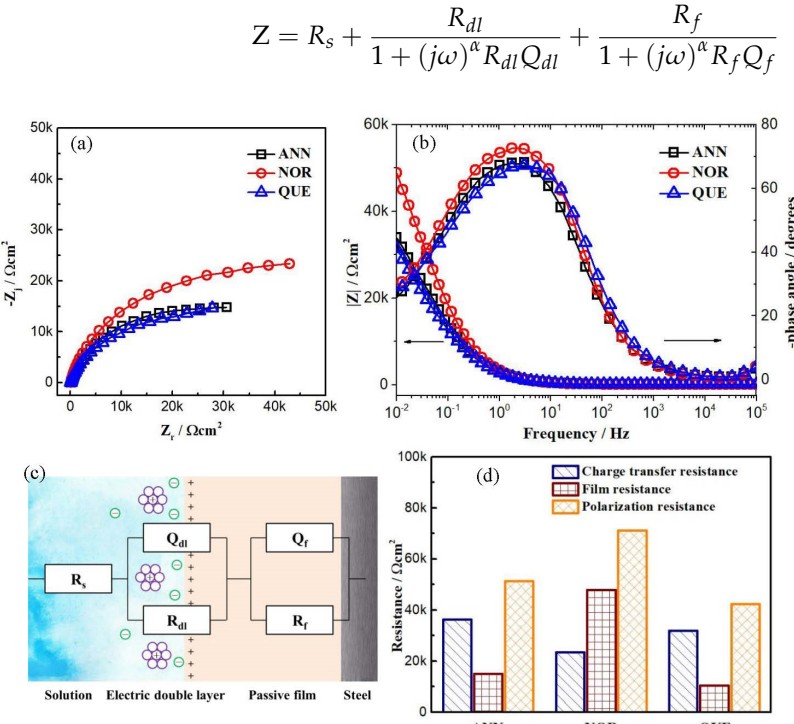

**Figure 7.** The electrochemical impedance spectroscopy for Cr-bearing steel in the saturated Ca(OH)$_2$ solution: (**a**) Nyquist plots, (**b**) Bode plots, (**c**) equivalent circuit, (**d**) fitted parameters.

The time constants for the medium and low frequencies are assigned to the charge transfer process in the double layer ($R_{dl}$-$Q_{dl}$ couple) and the redox reaction in the passive film ($R_f$-$Q_f$ couple), respectively. As shown in Table 3 and Figure 7d, the polarization resistance and film resistance can be established to follow the same order in terms of the passivation ability, with NOR > ANN > QUE, indicating that the dissolution rate of

the passive film on NOR is the slowest. In particular, the transport process for redox species ($OH^-/O_2$ or $Fe_3O_4/FeOOH$) at the pores on the film/solution interface is greatly suppressed for NOR, as the NOR film resistance is increased by 3.2 and 4.5 times compared to that for ANN and QUE, respectively. The results of electric resistances are consistent with the conclusions made from the XPS and MS measurements, demonstrating the formation of a compact and protective passive film on the surface of NOR. However, the charge transfer resistance for NOR is slightly decreased by 1.3–1.5 times compared to that for ANN and QUE.

**Table 3.** Fitting results for the impedance spectra.

| Sample | $R_s$ $\Omega cm^2$ | $Q_{dl}$ $Fcm^{-2}s^{n-1}$ | $n_{dl}$ $s^n$ | $R_{dl}$ $\Omega cm^2$ | $Q_f$ $Fcm^{-2}s^{n-1}$ | $n_f$ $s^n$ | $R_f$ $\Omega cm^2$ |
|---|---|---|---|---|---|---|---|
| ANN | $118 \pm 23$ | $2.50 \pm 0.08 \times 10^{-4}$ | 0.77 | $3.64 \pm 0.07 \times 10^6$ | $1.12 \pm 0.02 \times 10^{-4}$ | 0.85 | $1.50 \pm 0.09 \times 10^4$ |
| NOR | $112 \pm 10$ | $7.97 \pm 0.01 \times 10^{-4}$ | 0.87 | $2.35 \pm 0.01 \times 10^6$ | $2.54 \pm 0.05 \times 10^{-4}$ | 0.88 | $4.78 \pm 0.03 \times 10^4$ |
| QUE | $90 \pm 17$ | $2.42 \pm 0.14 \times 10^{-4}$ | 0.83 | $3.19 \pm 0.10 \times 10^6$ | $1.07 \pm 0.02 \times 10^{-4}$ | 0.83 | $1.05 \pm 0.11 \times 10^4$ |

### 3.3. Depassivation and Corrosion

Figure 8 shows the open-circuit potential evolution for Cr-bearing steels with respect to the incremental addition of chloride ions. At the initial stage of immersion, the OCPs range between $-360$ and $-300$ mV$_{SCE}$, which is lower than the empirical value of $-200$ mV$_{SCE}$ obtained in an actual concrete structure. These rather low potentials may be attributed to the low oxygen availability under the full immersion condition. Upon the gradual addition of chloride to the saturated $Ca(OH)_2$ solution, the OCPs change little initially when the chloride contents are relatively low. In fact, the passive film on steel transitions slowly from an ordered state to an amorphous-like oxide structure with increasing chloride content before breakdown [37], even though the corrosion potential appears to remain unchanged. The sharp shift in the OCP in the negative direction indicates a passivity breakdown for the steel, and the corresponding chloride content is defined as the chloride threshold [42]. The chloride threshold is sequenced as NOR (critical $[Cl^-]/[OH^-]$ of 43.94, critical $Cl^-$ content of 3.37 wt.%) > ANN (critical $[Cl^-]/[OH^-]$ of 20.81, critical $Cl^-$ content of 1.60 wt.%) > QUE (critical $[Cl^-]/[OH^-]$ of 6.94, critical $Cl^-$ content of 0.53 wt.%), suggesting that the breakdown of the passive film on NOR requires significantly higher chloride levels, while QUE is much more susceptible to chloride-induced corrosion.

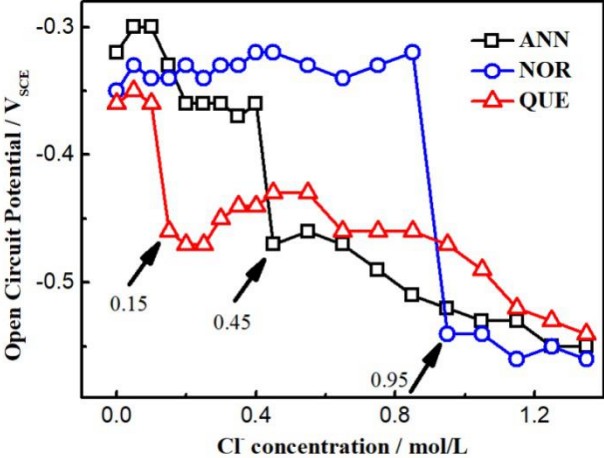

**Figure 8.** OCPs' evolution for Cr-bearing steels in the saturated $Ca(OH)_2$ solution with incremental addition of chloride ions.

Figure 9 shows the potentiodynamic polarization curves for Cr-bearing steels in a saturated $Ca(OH)_2$ solution with 1.25 M chloride ions. The corrosion processes include

the cathodic reaction of oxygen reduction and the anodic reaction of iron dissolution. The differences in anodic current density corresponding to a particular overpotential ($\eta$) among the three samples are much more notable than those for the cathodic current density, indicating that the heat treatment mainly influences the anodic process. The Tafel fitting results for the polarization curves are listed in Table 4. The significantly higher pitting potential ($E_{pit}$) for NOR suggests the formation of a more protective passive film against chloride penetration, which is consistent with the chloride threshold results. Table 4 also shows that NOR exhibits the lowest corrosion current density and the highest anodic Tafel slopes. The corrosion rate for NOR is approximately 1.61 times lower than that obtained for QUE. However, the difference in the corrosion rate is less considerable, indicating that the effect of heat treatment on corrosion propagation is less evident than on corrosion initiation.

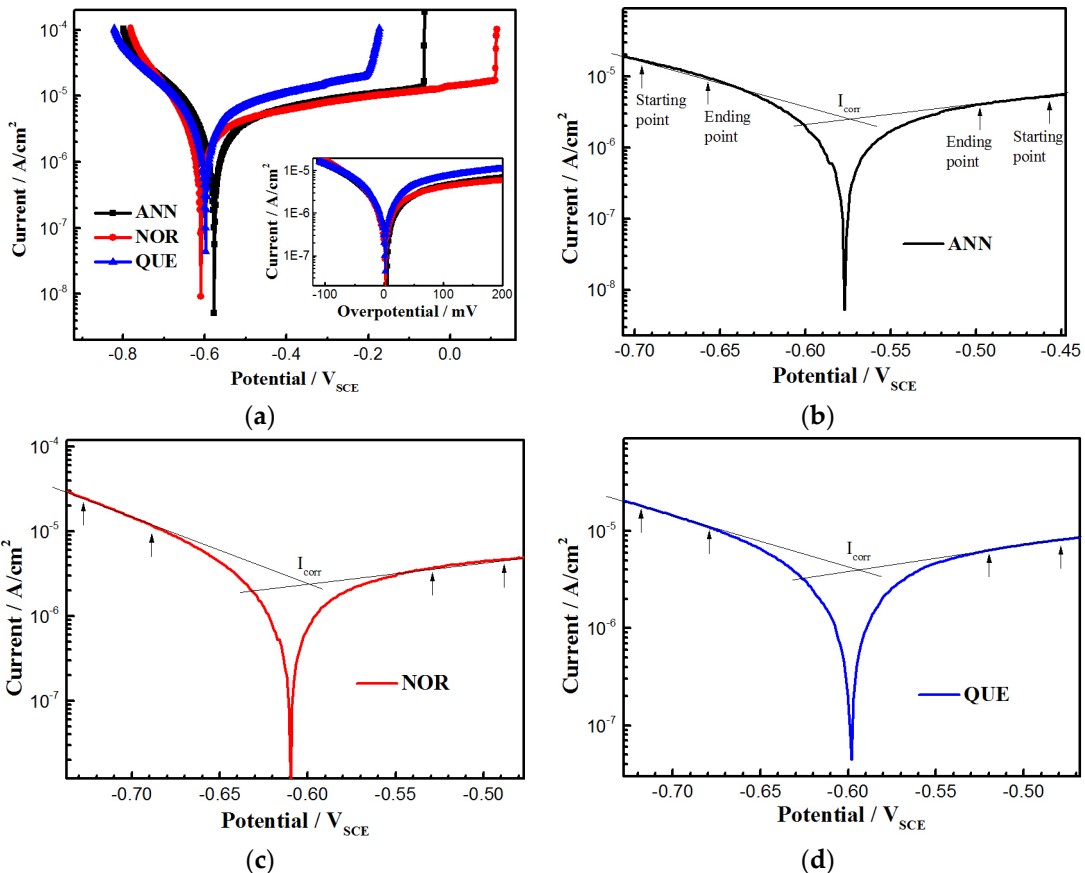

**Figure 9.** (**a**) Potentiodynamic polarization curves for Cr-bearing steels in the saturated Ca(OH)$_2$ solution with 1.25 M NaCl. (**b**) The fitting curves for potentiodynamic polarization curves of ANN (**b**), NOR (**c**), and QUE (**d**).

**Table 4.** Tafel fitting results of the potentiodynamic polarization curves.

| Sample | $E_{corr}$ mV$_{SCE}$ | $E_{pit}$ mV$_{SCE}$ | $\beta_c$ mV/dec | $\beta_a$ mV/dec | $I_{corr}$ µAcm$^{-2}$ |
|---|---|---|---|---|---|
| ANN | $-580 \pm 5$ | $-70 \pm 21$ | $161.85 \pm 6.38$ | $386.86 \pm 1.02$ | $2.72 \pm 0.05$ |
| NOR | $-610 \pm 10$ | $110 \pm 6$ | $120.77 \pm 9.51$ | $410.9 \pm 7.55$ | $2.44 \pm 0.10$ |
| QUE | $-600 \pm 8$ | $-200 \pm 37$ | $176.46 \pm 9.77$ | $376.19 \pm 6.90$ | $3.92 \pm 0.07$ |

Figure 10 shows the corrosion weight loss and corrosion morphology of steel samples after 10 days immersion in the saturated Ca(OH)$_2$ with 1.25 M NaCl. The dry-wet cycle experiment was chosen to simulate the extremely aggressive environment. The corrosion attacks are localized, and strong local corrosion occurred on the QUE sample. NOR exhibits

the lowest corrosion rate, which is approximately 56% of QUE. The weight loss is in line with the results from potentiodynamic polarization curves.

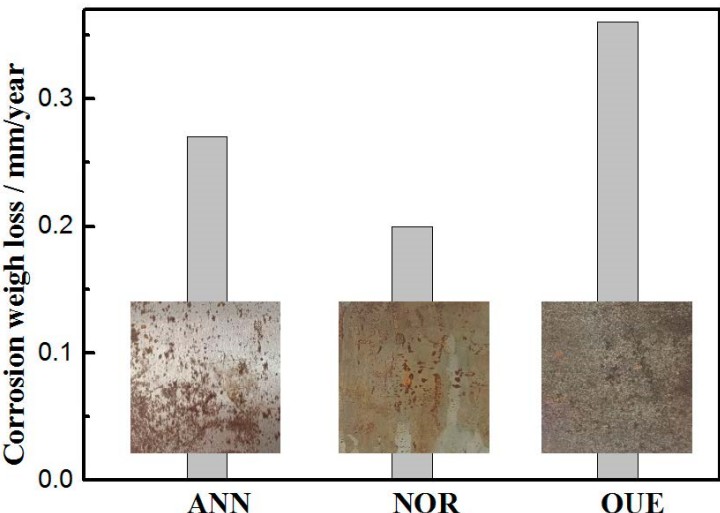

**Figure 10.** Corrosion weight loss and corrosion morphology for Cr-bearing steels in the saturated Ca(OH)$_2$ solution with 1.25 M NaCl.

## 4. Discussion

Heat treatment of quenching and tempering is commonly applied for steel reinforcement in order to optimize its mechanical properties [43]. However, according to the findings presented in Section 3, quenching and tempering can have adverse effects on the corrosion resistance of Cr-bearing steel in alkaline environments with chlorides. Specifically, after undergoing quenching and tempering heat treatment, the Cr-bearing steel reinforcement displays the poorest corrosion resistance, with the lowest critical [Cl$^-$]/[OH$^-$] and the highest corrosion rate. In contrast, normalization resulted in the best corrosion resistance for the samples.

From Figures 3 and 4, the main distinguishing factor in the microstructure for the three types of steel samples is the grain size. Generally, grain refinement is beneficial in terms of encouraging passivation and the formation of passive films. This is due to the high energy state and fast diffusion coefficient of the intercrystalline region. Ralston et al. [17] proposed an equation to model the correlation between corrosion rate and grain size for passive metals (corrosion rate < 10 μAcm$^{-2}$):

$$I_{corr} = A + B \times gs^{-0.5}$$

where A and B are constants, I$_{corr}$ is the corrosion rate (μAcm$^{-2}$), and gs is the grain size (μm). In the present study, it is observed that the corrosion resistance does not always increase with decreasing grain size under alkaline conditions, which contradicts the commonly accepted notion. For ANN, NOR, and QUE, the grain sizes are 19, 7, and 4 μm, respectively, their corresponding critical [Cl$^-$]/[OH$^-$] values are 20.81, 43.94, and 6.94, respectively, and the corrosion rates are 2.72, 2.44, and 3.92 μAcm$^{-2}$, respectively. That is, even though the specimen after quenching and tempering shows a fine-grained microstructure, its corrosion resistance remains suboptimal.

Theoretically, this phenomenon can likely be attributed to the redistribution of substitutional alloying elements in the alloy steel after heat treatment. The Cr-bearing steel, under thermodynamic equilibrium conditions, experiences significant enrichment of Cr at the cementite/matrix interface in the cementite, with a corresponding depletion in the matrix surrounding the carbide. However, when both a high degree of supercooling (quenching) and a low degree of tempering are present, the Cr enrichment process in the cementite is hindered during carbide precipitation from the supersaturated ferrite [44]. The different

partitions of Cr in ferrite and cementite on the phase and grain scales can lead to remarkable differences in the characteristics and protective properties of passive films.

However, the present work does not address experimental studies at the atomic scale, and, therefore, it is uncertain how heat treatment specifically affects the Cr element segregation and thus the protective properties of passive film. Relevant experiments, such as atomic force microscopy, Auger electron spectroscopy, and simulation calculations, will be further explored in subsequent research.

**5. Conclusions**

Low-alloy steels containing Cr are often utilized instead of ordinary carbon steel in coastal concrete structures to improve the durability of reinforced concrete. In some complex, high-stress concrete environments, the Cr-bearing steel undergoes heat treatment to enhance its strength, with quenching and tempering methods being the most widely employed due to their ability to refine grain and improve mechanical properties. The present research indicates that the Cr-bearing steel reinforcement exhibits microstructures with pearlite and ferrite after undergoing various heat treatments, including annealing, normalizing, and quenching and tempering. The grain sizes differ significantly among the three sample types, with values of 19 μm for ANN, 7 μm for NOR, and 4 μm for QUE. Furthermore, the Vickers hardness results indicate considerable variation, with respective values of 110 HV for ANN, 161 HV for NOR, and 249 HV for QUE.

Based on this study, although quenching and tempering improve the mechanical properties of Cr-bearing steel, their corrosion resistance significantly decreases in an alkaline environment containing chloride. On one hand, after quenching and tempering, the passive film offers the least protection for Cr-bearing steel, while it offers the most protection after normalization. The Cr/Fe and $Fe^{II}/Fe^{III}$ ratios in the passive film formed on QUE are much lower than those in NOR. Additionally, the space-charge layer of the passive film in QUE is much thinner, with a much higher defect density, and the film resistance is almost five times lower than that of NOR. On the other hand, the resistance to chloride-induced depassivation is lower for Cr-bearing steel after quenching and tempering than after normalization. The critical $[Cl^-]/[OH^-]$ value for QUE is approximately 6.6 times lower than that for NOR, and the corrosion rate is 1.6 times higher. Therefore, when designing or selecting the processes of heat treatment, the comprehensive effects of heat treatment on the mechanical and corrosion properties of low-alloy steel should be considered.

**Author Contributions:** Conceptualization, Y.T. and C.W.; methodology, Y.T. and D.Y.; validation, C.W., X.X. and P.D.; formal analysis, Y.T. and X.X.; data curation, D.Y.; writing—original draft preparation, Y.T.; writing—review and editing, C.W.; funding acquisition, Y.T. All authors have read and agreed to the published version of the manuscript.

**Funding:** This research was funded by [Natural Science Foundation of Guangdong Province] grant number [2021A1515012129] and [Science and technology program of Zhanjiang] grant number [2021E05005].

**Institutional Review Board Statement:** Not applicable.

**Informed Consent Statement:** Not applicable.

**Data Availability Statement:** The datasets generated and/or analyzed during the current study are available from the corresponding author on reasonable request.

**Conflicts of Interest:** The authors declare no conflict of interest.

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
