# Peer review of "Effect of Heat Treatment on the Passive Film and Depassivation Behavior of Cr-Bearing Steel Reinforcement in an Alkaline Environment"

_coatings, doi:10.3390/coatings13050964_

Round 1

Reviewer 1 Report

The article discusses the effect of heat treatment of Cr-bearing steel on its corrosion behavior. The weak point of the article is that the authors use only electrochemical methods to assess the corrosion rate of the steel under study. In order to increase the scientific and practical significance of the study, the authors need to perform gravimetric corrosion studies (to calculate corrosion weight and corrosion rate). The authors study the protective properties of the formed passive film, therefore, for comparison, it is necessary to provide data on the corrosion rate of steel without heat treatment. Also, for comparison, it is necessary to show SEM images and XPS data of polished steel in order to evaluate the change in the chemical composition of the surface as a result of heat treatment. To evaluate the effectiveness of heat treatment, authors should provide SEM images of steel samples after long-term corrosion studies.

The results of the electrochemical impedance spectroscopy are poorly described. Please, provide the equivalent circuit used and the results of the spectra fitting.

Please, provide the standard deviation data for all measurements.

Reviewer 2 Report

In this manuscript authors have presented the "Effect of heat treatment on the passive film and depassivation  behavior for Cr-bearing steel reinforcement in the alkaline environment".

(a) In fig. 2(g), recheck the average grain size as it is not matched with the bar diagram.

(b) The reference should be provided for equations.

(c) The Tafel fitting  results for the polarization curves are listed in Table 2. Authors need to show separate curve of fitting or in the inset of Fig 7.

(d) A comparison table of previously reported work will be appreciated in the introduction section.

Reviewer 3 Report

1.      Quantitative results need to be added in the abstract section.

2.      Please give a "take-home" message as the conclusion of your abstract.

3.      Keywords should be rearranged alphabetically.

4.      It is suggested to not use abbreviations in the keywords.

5.      The novelty in the current article by the authors is too weak. The past has seen extensive published work of written material. It is required to provide more details for more explanation about the present novel in the introductory section.

6.      To underline the submitted article gaps that the newest works tries to fill, it is crucial to explain the merits, novelty, and limits of earlier studies in the introduction.

7.      Apart from explerimental testing performed in the present study, The authors needs to explain potential further study adopting computational simulation of metals materials. It bring several advantages compared to experimental testing such as lower cost and faster results. Also, please refer the relevant reference as follows: Tresca Stress Study of CoCrMo-on-CoCrMo Bearings Based on Body Mass Index Using 2D Computational Model. Jurnal Tribologi 2022, 33, 31–8. https://jurnaltribologi.mytribos.org/v33/JT-33-31-38.pdf

8.      I am encouraging the authors for making a work's objective more clear.

9.      Please provide an additional figure in the introduction section in related submission work to improve the reader's understanding.

Round 2

Reviewer 1 Report

The paper can be accepted without any further changes.

Author Response

Thank you so much.

Reviewer 3 Report

1.      To enhance the understandability of the section on materials and methods easier for them to understand rather than just depending on the main text as it exists at the moment, the authors could add additional illustrations in the form of figures that explain the workflow of the present study.

2.      More information about tools, such as the producer, country, and specifications, should be included.

3.      The error and tolerance of the experimental tools used in this investigation are important aspects that have to be mentioned in the manuscript. It might be valuable for further research by other scholars because of the different results.

4.      Extend the information that chromium have been widely used in several mechanical application, one of the is for bearing of medical implant. Refer the relevant reference as follows: Tresca Stress Simulation of Metal-on-Metal Total Hip Arthroplasty during Normal Walking Activity. Materials (Basel). 2021, 14, 7554. https://doi.org/10.3390/ma14247554

5.      Results must be compared to similar past research.

6.      The discussion presented in this article is poor. A major improvement is mandatory, especially in discussing the results, not just simply mentioning them with a brief explanation.

7.      The limitation of the present submission needs to be added at the end of the discussion section before entering the conclusion section.

8.      Build the conclusion as a paragraph, instead of point by point as in the existing form.

9.      Further research should be discussed in the conclusion section.

10.   The authors sometimes reduced a paragraph to just one or two phrases across the whole article, which made the explanation difficult to follow. To make a more thorough paragraph, the writers should expand upon their explanation. It is advised to include at least three sentences in a paragraph, one of which should serve as the primary idea and the others as supporting details.

11.   The authors need to enrich the reference from five years back. MDPI reference is strongly recommended.

12.   The authors are encouraged to reduce their self-citation.

13.   Due to grammatical problems and linguistic style, the authors should proofread the work.

14.   After revision, provide a graphical abstract for submission.

Author Response

First of all, I would like to thank the reviewer for his/her constructive comments, especially points 5-10.

We have revised the article as per your request to the best of our ability. We hope the revised draft meets your requirements.
